# Past and future influence of climate change on spatially heterogeneous vegetation activity in China

Jiangbo Gao<sup>1,\*</sup>, Kewei Jiao<sup>1,2,\*</sup>, Shaohong Wu<sup>1</sup>, Danyang Ma<sup>1,2</sup>, Dongsheng Zhao<sup>1</sup>, Yunhe Yin<sup>1</sup>, and Erfu Dai<sup>1</sup>

<sup>1</sup>Key Laboratory of Land Surface Pattern and Simulation, Institute of Geographic Sciences and Natural Resources Research, Chinese Academy of Sciences, Beijing 100101, China,

<sup>2</sup> University of Chinese Academy of Sciences, Beijing 100049, China

\*These authors contributed equally to this work.

Correspondence to: Jiangbo Gao (gaojiangbo@igsnrr.ac.cn)

- Abstract. Climate change is a major driver of vegetation activity, and thus their complex relationships become a frontier and difficulty in global change research. In this paper, the spatial distribution and dynamic variation of climate change impacts on vegetation activity from 1980s to 2050 in China were investigated by the Geographically Weighted Regression (GWR) model, based on the combined datasets of satellite vegetation index, climate observation and projection, and future vegetation productivity simulation. Our results noted that the significant positive precipitation-vegetation relationship was
- and will be almost distributed in the north of China, except the northeast and northwest China with significant but varying influence of temperature rising, while the regions with temperature dominated vegetation activity mainly located in the southern part of China. There will be different climatic dominators for vegetation activity in some regions such as northwest China, and even opposite correlation in the northeast China, and further the responding patterns of vegetation activity to precipitation variation will be moving southward in the future three decades. It is indicated that although high warming rate
- will restrain the vegetation activity, precipitation variability can mediate the hydrothermal conditions for vegetation activity, for example the enhanced vegetation activity in the Tibetan Plateau and the weakened vegetation activity in the East and Middle China in the future. Furthermore, coupling the responding patterns and the dynamic variation, it can be found that during the period from 2021 to 2050, vegetation in most of north China may adapt to an arid environment, while in many southern parts it will be repressed due to water shortage. However, the continuous and dynamic responding process of
- vegetation activity to climate change will be determined by the spatial heterogeneity in climate change and vegetation cover.

### **1** Introduction

Vegetation is the primary producer of terrestrial ecosystems, and also plays a significant role in the energy transfer between land and atmosphere and the climate regulation (Wang et al., 2012). Many features, such as vegetation growth, coverage, phenology and carbon cycle process, can mirror the vegetation activity at large scales (Fang et al., 2004). As the necessary environmental variables, climate change has and will have profound influences on the spatially heterogeneous vegetation

activity (Zhao and Running, 2011; Dong and Sutton, 2015). Investigation on the vegetation response to climate change and its spatial variation can provide theoretical basis for ecosystem-based adaptation, and thus becomes a major topic of global change research.

- Studies on the relationships between vegetation activity and climate change have been carried out at different spatial scales (Krishnaswamy et al., 2014; Seddon et al., 2016). However, the specific variation in vegetation activity depends on different environmental conditions and their changes. Del Grosso et al. (2008) found that 13 % increase in total global Net Primary Productivity (NPP) for potential vegetation over the last century due to a significantly positive correlation between NPP and climatic factors. As for the regional scales, a continuously rising trend of NPP at a rate of 0.22 %/a was noted in the tropical and subtropical zones, while a continuously decreasing trend (-0.05 %/a) appeared in the temperate zone in China
- 10 (Wang et al., 2008). Additionally, because of the spatial heterogeneity in vegetation cover, together with the strong regional differences in precipitation (de Jong et al., 2013; Zeppel et al., 2014; Duo et al., 2016), the increased precipitation concentration caused the decreased NPP in semiarid regions (Fay et al., 2008; Hoover et al., 2014), but increased NPP in arid regions (Baez et al., 2013).
- Most former research applied correlation and trend analysis to detect vegetation effects of climate change, but studies on the dynamic response of vegetation activity to climate variability and the influencing mechanism were still relatively weak (Reyer et al., 2013). Actually, the relationship between climate and vegetation may vary with time scales, including years, seasons, and even day and night (Piao et al., 2008, 2014; Peng et al., 2013). For example, Wang et al. (2016) found a large percentage increase of NPP in northwest China and Tibetan Plateau over the 21st century under the four emission scenarios, which was different from the decrease trend over the past. Thus, due to the crossing effects of spatial and temporal scales, the vegetation-climate correlations present the characteristics of complexity and uncertainty.

Generally, previous studies on climate-vegetation relationships in China were still concerned with historical periods and local features (Wang et al., 2005; Baez et al., 2013). Transfers of responding patterns from past to future at national scale need to attract more attention, or it is not conducive to the development of targeted, orderly adaptation strategy. Therefore, in this paper, the geographically weighted regression (GWR) and other statistic methods were used to investigate the response

25 pat

patterns of vegetation activity to climatic change and their variability from 1980s to 2050 in China, in order to identify the effective factors and regions during different periods. And further, through the dynamic description of climate induced vegetation activity from rise to decline in different regions, it is expected to provide scientific basis for making ecosystem-based adaptation strategies in response to global climate change.

### 2 Materials and methods

### 2.1 Data

### 2.1.1 Vegetation activity

Two vegetation indicators, including Normalized Difference Vegetation Index (NDVI) and Net Primary Productivity (NPP), were applied to present vegetation activity (Robertson et al., 2009; Wright et al., 2012). The satellite-based NDVI is independent from the climate observations, so they are appropriate for correlation fitting. As for the future scenario, the projected NPP simulated by ecological model was applied to analyze the climate-vegetation relationships.

Earth System

Dynamics

Discussions

NDVI dataset from 1982 to 2013 at a spatial resolution of 8 km and temporal interval of 15-day were obtained from the GIMMS (Global Inventory Modeling and Mapping Studies) group, and then the original NDVI dataset was resampled to 0.5 separately resolution by AraCIS 10.3 to match alignets dataset. This dataset is because a its high available and the because of the

10 0.5 °spatial resolution by ArcGIS 10.3 to match climate dataset. This dataset is known as its high quality and the longest time series, and has been widely used in many studies of global and regional vegetation activity (Tucker et al., 2005; Mao et al., 2012).

Projected NPP from 2021 to 2050 at a spatial resolution of 0.5 ° were simulated by LPJ-DGVM (Lund-Potsdam-Jena Dynamic Global Vegetation Model). Climate data imported to the model was divided into two parts: one was derived from the historical simulation data (1981-2010) published by the National Climate Center of CMA (China Meteorological Administration) in order to validate the climate simulation results, and the other was climate scenario data from 2021 to 2050 under RCP4.5, a stabilization emission scenario that assumed the imposition of emissions mitigation policies. And the soil texture data imported was derived from FAO (Food and Agriculture Organization of the United Nations), with a 1 ° spatial resolution. The detailed information was described in the following parts.

### 20 2.1.2 Climatic variables

In the past analysis, raster data of 0.5 ° spatial resolution, which were interpolated from 2472 meteorological stations nationwide through thin plate spline (TPS) combining with 3D geospatial information by the National Climate Center of CMA, was applied to represent climate change, including annual temperature and precipitation from 1982 to 2013. In the future analysis, monthly climatic data including temperature and precipitation from 2021 to 2050 were provided by the

25 National Climate Center of CMA. They simulated these data of China under RCP4.5 emission scenario by regional climate mode (RegCM4.0) from the Abdus Salam International Centre for Theoretical Physics (ICTP), and interpolated them into 0.5 °spatial resolution.

### 2.2 Methods

### 2.2.1 Lund-Potsdam-Jena Dynamic Global Vegetation Model (LPJ-DGVM)

The LPJ-DGVM has been widely applied to study the vegetation dynamics including the processes of establishment, mortality and disturbance by wildfires (Sitch et al., 2003). The input data for LPJ-DGVM included the climate data (i.e. monthly temperature, precipitation and cloud cover), soil texture data, and atmospheric CO<sub>2</sub> concentration. Based on the photosynthesis, canopy energy balance, distribution of photosynthates in plants and soil water balance, the model could simulate the stomatal conductance, photosynthesis, respiration, foliage and leaf litter, resource competition, tissue turnover, soil microbial decomposition process, and then calculate the carbon cycle, CO<sub>2</sub> and moisture flux, photosynthetic rate, primary productivity and carbon storage of vegetation. Moreover, variation of plant functional types (PFTs) was taken into

- 10 account in the simulation. At first, a 1000-year model spin-up was run in order to approach equilibrium with respect to vegetation covers and carbon pools for the terrestrial ecosystem. And then, LPJ-DGVM was driven by climate data (Sect. 2.1.1) to execute a simulation for projected NPP. For better simulation in China, LPJ-DGVM was carefully improved by adding shrub and cold grass PFTs, which were parameterized based on various inventory and observational data in accordance with the characteristics of ecosystems in China (Zhao et al., 2013). The simulated NPP by the modified model
- 15 was validated with the data of observed sites in China, where the correlation coefficient ( $R^2=0.64$ , p

10

20

### 2.2.2 Geographically weighted regression (GWR)

GWR extends the traditional regression techniques to consider the spatial heterogeneity in climate-vegetation correlations by assigning weight values (Fotheringham et al., 2002). So the regression and its parameters in each point of space are quantified separately and independently. It was conducted to reveal the spatial variations in relationships between vegetation

5 and climatic variables. Both the spatial distribution and the dynamics patterns were considered by the GWR model. It is represented as:

$$y_{i} = \beta_{0}(\mu_{i}, \nu_{i}) + \sum_{k=1}^{p} \beta_{k}(\mu_{i}, \nu_{i}) x_{ik} + \varepsilon_{i}, \qquad (1)$$

where  $y_i$  is dependent variable,  $x_{ik}$  is independent variables,  $\varepsilon_i$  represent the random error term. ( $\mu_i$ ,  $v_i$ ) expresses the spatial location of the sample *i*. *k* denotes the independent variable number.  $\beta_0$  is the intercept and  $\beta_k$  is the regression parameters at location *i*. In this paper, the multi-annual average climatic variables and vegetation activity were applied in GWR for getting responding patterns, and further their annual variability was used to detect dynamic response.

The regression coefficients are expressed by:

$$\beta(\mu_i, \nu_i) = (X^T W(\mu_i, \nu_i) X)^{-1} X^T W(\mu_i, \nu_i) Y,$$
<sup>(2)</sup>

where  $\beta(\mu_i, v_i)$  represents the estimate of the regression coefficient.  $W(\mu_i, v_i)$  is the weighting matrix, and *X* and *Y* are 15 matrices for independent and dependent variables, respectively. The weighting matrix can be calculated as follows:

$$\omega_{ij} = \exp(-\frac{d_{ij}^2}{b^2}),\tag{3}$$

where  $\omega_{ij}$  expresses the weight of sample *j* for sample *i*,  $d_{ij}$  represents the Euclidean distance between samples *i* and *j*, and *b* is the kernel bandwidth. Furthermore, the corrected Akaike Information Criterion (AICc) and the coefficient of determination (R<sup>2</sup>), describing model's predictive ability, were performed to determine the appropriate bandwidth, meaning lower AICc and higher R<sup>2</sup>.

### 2.2.3 Trend analysis

The vegetation and climatic trend for each pixel was estimated by ordinary least squares (OLS) based on unitary linear regression using ArcGIS 10.3 software:

$$\theta_{slope} = \frac{n \times \sum_{i=1}^{n} i \times Y_i - \sum_{i=1}^{n} i \sum_{i=1}^{n} Y_i}{n \times \sum_{i=1}^{n} i^2 - (\sum_{i=1}^{n} i)^2},$$
(4)

25 where  $\theta_{slope}$  is the regression slope that represents the change trend. *n* is the sequence number of years. *Y<sub>i</sub>* refers to the climatic or vegetation variables for the specific year.

### 2.2.4 Accumulative anomaly

Accumulative anomaly analysis was used to identify the turning year in the future changes of NPP in this study. It is a commonly method to identify the changing tendency of discrete data. The equation can be expressed as follows:

$$\widehat{X}_{i} = \sum_{i=1}^{n} (NPP_{i} - \overline{NPP}),$$

(5)

5 where  $X_i$  is accumulative anomaly for the NPP in the *i*th year.  $\overline{NPP}$  is the mean value of the  $NPP_i$ , and n is the sequence number of years. As shown in this formula, the accumulative anomaly can be used to analyze the fluctuation magnitude of a series of NPP. If the changing curve is comprised at least two parts of the tendency, then the turning year can be found.

### **3 Results**

Spatial regression method, GWR model in this study, was adopted to analyze vegetation activity in relation to climatic variables for the periods of 1980s-2010s and 2021-2050. Through simulating the climate-vegetation relationship and its conversion, the transfer traits of responding patterns in the future can be identified.

### 3.1 Existing climate change impacts on vegetation activity and their distribution

### 3.1.1 Climate-vegetation relationships: annual average

The normalized annual temperature, precipitation and NDVI, after the check of collinearity, were used to calculate the regression coefficient from GWR model. The normalized coefficients for climate-vegetation relationships were applied to reveal the relative importance of climate factors for vegetation activity in different regions of China. Combined with the analysis on vegetation and climate trends, the climatic determinants for vegetation activity can be obtained. At then, the 'effect region', which is defined as the spatial domain of climatic determinants for vegetation activity, has been illustrated in Fig. 1.

- 20 Temperature and precipitation played a promoting role on vegetation activity in most parts of China. The effect regions of positive precipitation-vegetation relationship were found in the north of China, such as southwest of the Northeast Plain and the Loess Plateau. While the positive temperature-vegetation relationship were distributed in the south of China, including the Huanghuai Plain, Qinling Mountains, Yangze Plain and the Western Sichuan Plateau to Yunnan-Guizhou Plateau.
- In the Greater Khingan Mountains, Sanjiang Plain and several parts of northwest China, the mainly type of effect regions was a negative temperature-vegetation relationship, which meant the vegetation activity weakened with warming. The evapotranspiration accelerated with temperature increasing, which lead to the drought (Zhang et al., 2014). In some regions like the Tibetan Plateau, the effect of temperature and precipitation was not consistent with the varying trend of NDVI. It is indicated that other factors (terrain, radiation, etc.) have more significant effect on vegetation than temperature
- 30 and precipitation.

### 3.1.2 Climate-vegetation relationships: annual variability

Combined with the analysis on vegetation and climate trends (Fig. 2), the climatic determinants for vegetation dynamics can be obtained. The climate variability and the varying rate of NDVI were applied in GWR as independent and dependent variables, respectively, to conduct the spatial correlation analysis on climate change and vegetation dynamics.

- 5 The correlation between NDVI variability and temperature variability was positive in the areas of the North China, the northwest and southeast of China (Fig. 3a), where NDVI rate was increasing (Fig. 2a). Among these areas, the North China and northwestern part were located in the effect regions of positive precipitation-vegetation relationship (Fig. 1). The precipitation was increasing (Fig. 2b) and the correlation between NDVI variability and precipitation variability was positive (Fig. 3b). The southeastern part was located in the effect regions of positive temperature-vegetation relationship. However, the precipitation trend and vegetation dynamic response to precipitation variability were opposite to the former. Therefore, 10

both temperature and precipitation variability influence the vegetation activity together.

In the Loess Plateau, the East China, and the southwest and northeast of China, correlation between NDVI variability and temperature variability was negative (Fig. 3a). The Loess Plateau, where precipitation was increasing (Fig. 2b), was located in the effect regions of positive precipitation-vegetation relationship (Fig. 1), and the correlation between NDVI

- 15 variability and precipitation variability was positive (Fig. 3b). The fastest decrease of NDVI rate was distributed in the northeast of China (Fig. 2a). On one hand, the dynamics response to temperature variability was negative there, which was located in the effect regions of negative temperature-vegetation relationship. On the other hand, the precipitation trend in this area was increasing. But the dynamic response patterns to precipitation variability were discontinuous.
- According to the upper analysis, it is indicated that the effect of hydrothermal combination on vegetation activity is 20 very important in the context of warming. The high warming rate will lead to excessive evapotranspiration of vegetation and soil moisture, which may further cause to drought and restrain the vegetation growth. Meanwhile, most of the variations in precipitation rate can adjust the hydrothermal conditions for vegetation growth and promote its activity to a certain extent. Moreover, when the dynamics response to climatic variability is consistent with the effect regions, this kind of climatic factor and its variability will play an important role on the vegetation activity, combined Fig. 1 with Fig. 3.

#### 3.2 Future scenarios of climate-vegetation correlations and the varying patterns 25

### 3.2.1 Future responding patterns for annual average of vegetation activity

As the temperature rises, the vegetation response to temperature is more and more evident in the Tibetan Plateau, and the effect regions will turn to a positive temperature-vegetation correlation, the eastern part of which in the past will be moving southward (Fig. 4a). In several parts of the Loess Plateau and Inner Mongolia Plateau, the effect regions will be changed

because the influence of positive temperature-vegetation correlation will gradually surpass that of precipitation. The 30 temperature response will transform from negative to positive correlation in the Sanjiang Plain-Changbai Mountain of northeast China, which indicates that the hydrothermal condition has been improved in these areas.

The effect regions of positive precipitation-vegetation correlation have been extending southward to the Hubei and Hunan Plain and Sichuan Basin, and in the south of 25 ° N the effect regions will be positive temperature-vegetation correlation (Fig. 4a). In the Junggar Basin the positive correlation of precipitation is weaker than the past and the effect regions will turn to a negative correlation of temperature, indicating the worsened drought condition in the future. In addition, the effect regions of others will be increasing in the central of the Northeast Plain and southeast of the North China Plain,

5

## which indicates that the effect regions will become more complicated in these areas in the future.

### 3.2.2 Future responding patterns for annual variability of vegetation activity

Compared with the past, the response to temperature variability will turn to a positive correlation in the Yunnan-Guizhou Plateau, Tibet Plateau and most areas of east China, while it will turn to negative in the Inner Mongolia and east Qinghai
province (Fig. 4b). However, with the complex interaction of water and heat, the spatiotemporal variation of precipitation in the future will be more intense, which makes the dynamic patterns of precipitation more fragmented (not shown).

Precipitation trend will be changing from a downward trend to upward from the Sichuan Basin to the east of Yunnan-Guizhou Plateau (Fig. 5b), where the effect regions will turn to a positive precipitation-vegetation relationship (Fig. 4a). In the east of Yunnan-Guizhou Plateau, the responses to temperature and precipitation variability are positive, which reveals

that both temperature and precipitation will be conducive to vegetation activity in these areas. But in the Sichuan Basin, the response to temperature variability is negative, so the increasing precipitation will play a greater role there on vegetation activity.

In contrast, precipitation trend from upward to downward has been mainly distributed in the North China Plain and the south coast (Fig. 5b). The reason for NPP decreasing in North China Plain (Fig. 5a) is that the precipitation reduction will

- lead to the drought aggravated. And the response to precipitation variability will turn to a negative relationship in most parts, which means that the faster the precipitation decreasing rate, the slower rate of NPP decreasing. It may indicate that the vegetation have been adapted to the warm and dry environment gradually. In the south coast, however, NPP will be continuous increasing there (Fig. 5a). The effect regions will turn to a positive temperature-vegetation relationship, which is consistent with the response to temperature variability. It is indicated that an increase in warming rate will play a greater role
- there. At the same time, the response to precipitation variability will be negative relationship in the future. That is, the faster the precipitation rate decreasing, the more rapid increased rate of NPP. Therefore, both temperature and precipitation variability will promote the vegetation activity together in these areas.

### 3.3 Response patterns and processes at different stages in the future

Furthermore, in order to discuss the process of vegetation response to climate change, the transfer of spatial patterns and its control mechanism from an uptrend period to a downtrend of NPP have been detected. Then we tried to solve the question whether the change of vegetation trend in different regions is transitional and explore the threshold when the trend will be varying.

### 3.3.1 Stages classification and spatial changes of effect regions

From the inter-annual variations of NPP in the next 30 years (Fig. 6a), it shows a more significant downward trend after the year of 2039 (Fig. 6b). Then the accumulative anomaly of NPP in 2021-2038 is illustrated in Fig. 6c. It is shown that there is an increasing trend prior to the year of 2030, and a decreasing trend after 2030. The inflexion point of the NPP change is

5 2030 in this period. Therefore, we selected an upward trend (T1:2021-2030) and a downward of NPP (T2:2039-2050) as two future periods in this paper.

The effect regions of climatic factors on vegetation in different periods are illustrated in Fig. 7, and its change areas are summarized in Table 2. Compared with T1, the positive correlation of precipitation will be more and more obvious near the middle and lower reaches of the Yangtze River in T2 period. However, the precipitation there will be decreasing dramatically during T2. So the increase of temperature will accentuate the importance of moisture, which is the main reason for the NPP decline there. The negative correlation with temperature has been reflected in the north China, where the increasing temperature will accelerate the evapotranspiration. Negative precipitation-vegetation correlation will begin appearing in the north of Greater Khingan Mountains and the east of Yunnan-Guizhou Plateau, in which the variation trend of precipitation is unevenly and the upward trend of NPP will be slightly weaker than T1. Moreover, the effect regions of

15 others will restrain the vegetation activity in the Jiangnan Hills, Sichuan Basin and the Western Sichuan Plateau, which indicates that the effect regions will become more complicated.

For the dynamics responding patterns, the negative correlation with temperature variability will almost extend nationwide. It can be seen that the warming rate will become the main reason for inhibiting vegetation activity. Precipitation variability will still reflect the characteristics of uneven distribution. Negative correlation regions will increase in northern

parts of China, in which the applicability of vegetation to water shortage environment will be enhanced. In the southern parts, however, the negative correlation of precipitation variability will have been reduced. Warm and dry environments will continue to limit vegetation activity.

### 3.3.2 Quantitative detection on variation in vegetation trends

- Compared with precipitation, the temperature will be changing regularly along with the vegetation trend, which clearly 25 reflects its transformation. Theoretically the NPP response is a bell-shaped with temperature change (Fig. 8). Therefore, the NPP trends in T1 and T2 were compared and 6 regions have been divided based on the difference (Fig. 9). Changes in temperature, precipitation and their variability in different areas have been given in Table3. R1, where NPP will transform from rising to declining with more warming ( $0.70 \pm 0.29$ °C), is mainly distributed in the Northeast and North China Plain. R3 is mainly distributed in the south of China and the upward trend of NPP there will have slowed down. R6, indicating that
- vegetation activity will adapt to the high temperature and low precipitation, is widely distributed in the northwest arid regions. As is shown in Fig. 11, there is an obvious spatial continuity between different regions (R1and R3, R5; R6 and R5; R4 and R3 etc.). It is indicated that the trend of vegetation change is a dynamic and continuous process, which the

development order is R4-R3-R1-R5-R6 respectively. That is, the uptrend of NPP will slow down, enter a downward trend and begin to accelerate, and finally adapt to the harsh environments. When the climatic conditions improve, the vegetative activity may be restored to the previous state (R2).

In order to study the variation process of vegetation and the corresponding temperature rising, the average NPP and its 5 trend variation ( $\Delta NPP$  and  $\Delta Nt$ ) divided by the temperature range of 0.2 °C, were calculated and fitted by nonlinearly curves in the specified region (Fig. 10). It should be noted that  $\Delta NPP$  represents the variation of NPP and  $\Delta Nt$  expresses the variation of NPP variability from T1 to T2.

R4 and R3 occupy most regions of southwest China. NPP will continue to increase and  $\Delta NPP$  will first increase (R4) and then decrease (R3) as the temperature increases (Fig. 9). The increase of  $\Delta Nt$  will be from fast to slow and then decrease 10 after A2 ( $\Delta Nt = \frac{\partial \Delta NPP}{\partial \Delta T} = \frac{\partial^2 NPP}{\partial \Delta T^2} = 0$ ), which is the turning point during the process of R4 to R3 (Fig. 10a). It denotes that NPP uptrend rate begins to decrease when the temperature is raised about  $0.33^{\circ}C(\pm 0.01^{\circ}C)$  in the southwest. When the temperature keeps increasing more than A3(0.475 ± 0.01^{\circ}C), NPP may show a tendency to adapt to adverse environment, while NPP will be in the process of turning into a growth trend with the improvement of environmental conditions before A1.

15 first decrease (R5) and then increase (R6) (Fig. 9). The decrease of  $\Delta Nt$  will be from fast to slow and then increase after B  $(\Delta Nt = \frac{\partial \Delta NPP}{\partial \Delta T} = \frac{\partial^2 NPP}{\partial \Delta T^2} = 0)$ , which will be the turning point from R5 to R6 (Fig. 10b). It means that vegetation in the Loess Plateau will begin to improve the water use efficiency (WUE) in order to adapt to harsh environment if the temperature is raised about 0.43 °C (±0.01 °C).

Most parts of the Loess Plateau will be located in the change of R5 to R6. NPP will continue to decrease with  $\Delta NPP$ 

At last, NPP will be experiencing a process from increasing to decreasing in the northeast (R3-R1-R5),  $\triangle NPP$  will 20 continue to decrease and then turn into downtrend after C2 ( $\triangle NPP = \frac{\partial NPP}{\partial \Delta T} = 0$ ), which will be the tipping point of vegetation activity from rise to decline (Fig. 10c). That is, the vegetation trend will be reversed when the temperature will be raised more than 0.83 °C (± 0.01 °C) in the northeast and enter into the adaptation stage after the temperature is over 1.02 ± 0.01 °C (C3), while it reflects the trend of previous process (R4) before C1.

### 4 Discussion

### 25 4.1 Responding mechanism

From the past to future, it can be seen that the response of vegetation activity to climate change is more and more obvious, which is consistent with the research results in other areas (Piao et al., 2011; Urban, 2015). However, climate-vegetation relationship is complex with nonlinear characteristics. Specifically, before the optimum temperature for photosynthesis, a temperature rise will promote vegetation activity by an accelerated release of nutrients and improved availability from the

30 soil (Michaletz et al., 2014), such as the Tibetan Plateau in the future. When overpass this temperature, however, the

respiration will be promoted and accelerate the nutrients consumption, especially in the effect regions of negative temperature-vegetation relationship in the northwest of China. On the other hand, water shortage is not conducive to the transport of nutrients, thereby reducing the accumulation of organic matter (Brohan et al., 2006). So the precipitation variation can adjust the vegetation activity to a certain extent. However, vegetation in several northern parts may gradually adapt to the drought because the warming there will weaken the varying rate of photosynthesis and respiration, which cause the NPP downtrend slowed.

5

### 4.2 Spatial heterogeneity

In this study, the effect region was defined and applied to identify the dominated climatic factors for vegetation activity in different areas of China during the past and future 30 years. Corresponding to future global warming, the importance of available water for vegetation activity will be more prominent, which causes the effect regions of positive precipitation-vegetation relationship moving southward in the future. On the other hand, there may be different explanations for the same type of correlation in different regions, for example, in the Tibetan Plateau and parts of North China Plain, the rel