# Peer review of "Past and future influence of climate change on spatially heterogeneous vegetation activity in China"

_Earth System Dynamics, 2017_

## Referee Comment (RC1) · Anonymous Referee #1 · 1 Mar 2017

This manuscript tries to disentangle the impact of climate change on vegetation activity. Vegetation activity is either satellite derived NDVI for the past or model derived NPP for the future. In my opinion the manuscript does not meet a high enough scientific standard in its current form and I found it really hard reading the paper due to language deficits. Therefore I recommend to reject the manuscript. In the following are a few examples on what should be improved.

Scientific standard

1.) The validation promised on page 3 line 16 of past NPP versus NDVI is simply mentioned on the next page in the Methods section. This belongs to Results and needs to be proven.

2.) Section 2.1.2 Is there a reference for the past climate data? How does it compare

to established reconstructions e.g. CRU (Mitchell & Jones, 2005)?

3.) Page 3 line 14-16 The sentence is almost identical to a sentence of the previously mentioned Zhao et al. (2013) reference without referring to it.

4.) Page 4 line 21-22 "On one hand ... On the other hand ..." This is an interpretation and does not belong into the Methods section.

Language

I am also not a native English speaker like the authors and like a lot of potential readers. I must say, I did not understand most of the sentences in the manuscript. Therefore the manuscripts needs substantial language editing. Some examples:

1.) Page 1 line 16-17 "while ..." there is no verb in this sentence.

2.) Page 12 line 10-11: "While ..." This is just half a sentence.

3.) "temperature rising" throughout most of the manuscript should be rising temperature.

Others

In general the sentences are to long and there are too many commas in the wrong places. In the map figures there is a small pictogram of the "South China Sea Islands" without any new data in it, remove it.

Mitchell, T. D. and Jones, P. D.: An improved method of constructing a database of monthly climate observations and associated high-resolution grids, Int. J. Climatol., 25(6), 693-712, doi:10.1002/joc.1181, 2005.
* * *